# Role of marriage, motherhood, son preference on adolescent girls' and young women's empowerment: Evidence from a panel study in India

Lakshmi Gopalakrishnan [1¤]*, Stefano Bertozzi[1], Sophia Rabe-Hesketh[2]

**1** Department of Health Policy and Management, Berkeley, CA, United States of America, **2** School of Education, UC Berkeley, Berkeley, CA, Unites States of America

¤ Current address: Institute for Global Health Sciences, University of California, San Francisco, San Francisco, CA, Unites States of America

* lakshmi.gopalakrishnan2@ucsf.edu

## Abstract

### Background

Marriage is a key determinant of health and well-being of adolescent girls and young women (AGYW) in India. It is a key life event in which girls move to their marital households, often co-residing with their in-laws and begin childbearing. The change in the normative environment in conjunction with cultural norms surrounding son preference influences women's overall life course. However, there is scant research about the association between these life transitions and changes in empowerment among AGYW in India.

### Methods

Using two waves of data from prospective cohort panel dataset that followed unmarried (6,065 observations in each wave) and married AGYW (3,941 observations from each wave) over a three-year period from Uttar Pradesh and Bihar, we examined how marriage, childbearing, and having a son is associated with changes in AGYW's empowerment, especially considering whether AGYW marry into patrilocal households (household with in-laws) as an effect modifier. Empowerment indicators included freedom of movement or mobility, decision-making power, access to economic using Kabeer's framework as our theoretical approach.

### Results

Marriage was associated with lower freedom of movement with a pronounced effect on those who co-resided with their in-laws. Marriage was associated with greater decision-making power for AGYW who did not co-reside with the in-laws. Motherhood was positively correlated with greater freedom of movement, marginally higher intrahousehold decision-making power, and better access to economic resources. No statistically significant evidence that having at least one son compared to having daughters only (or no daughters)

**Data Availability Statement:** The data that support the findings of this study are openly available in Harvard Dataverse at https://doi.org/10.7910/DVN/RRXQNT (Council, 2018).

**Funding:** The authors received no specific funding for this work.

**Competing interests:** The authors have declared that no competing interests exist.

**Abbreviations:** UDAYA, Understanding Lives of Adolescents and Young Adults; AGYW, Adolescent Girls and Young Women.

conferred additional changes in girls' freedom of movement, intrahousehold decision-making power, and access to economic resources.

## Conclusion

Findings highlight the importance of understanding the vulnerabilities of being newly married in adolescence and emphasize the need for having interventions that target newly married AGYW along with mothers-in-law to empower them.

## Background

Marriage is one of the major determinants of health and well-being of adolescent girls and young women (AGYW) aged 15–24 years and their children in low-and-middle income countries [1, 2]. In patrilocal societies such as India, the cultural practice in which women move from a natal household to their marital household puts them in a new unfamiliar social and household environment [3].Women tend to lose ties with their natal kin and move to a household where they have subordinate status to both men and senior women of the household. Senior women, especially the mother-in-law, have control over household resources [4–8].

The next transition that affects young women is childbearing, which often comes immediately after marriage in South Asia, especially among lower socio-economic strata [9]. In India, the median age at first marriage is 19 years among women aged 20–49 years. 27% of women aged 20–24 years had a "child marriage," i.e., married before they were 18 years old [9]. Childbearing often begins soon after marriage—about 60% of women under 25 were pregnant within the first year of marriage. Women in India spend most of their lifetime in the institution of marriage [9]. About 50% pregnancies in India are unintended, with young women below 24 years having the highest rate of unintended pregnancies [9].

Evidence from India demonstrates that young married women often do not have decision-making power to use contraception [10] and little power to influence decisions on fertility reflected in 40% of young married women (15–19 years) having unmet need for family planning [9, 11]. Early childbearing is detrimental to women and their children's nutritional status, healthcare utilization, and iron levels [12–16]. Systematic review from low-and-middle-income countries also highlights the influence of adolescent childbearing on greater rates of morbidity and mortality with greater hypertensive disorders among young mothers compared to the older women [17].

Intertwined with giving birth in South Asia is gender bias at birth. Son preference is one example of deep-rooted culture-specific gender norms that result in health inequities for girls and women. Women are subjected to extreme social pressure to bear a son and undergo repeated pregnancies until a son is born [18–21]. In patrilocal-patrilineal societies, sons are perceived to have greater economic, social, and religious utility compared to daughters because sons (or other male descendants) carry forward the family's name and legacy and inherit family assets, perform religious duties, and co-reside with their elderly parents caring for them during their old age [18, 22]. A prior study that leveraged several rounds of India's Demographic Health Survey data suggested that mothers (15–49 years) who have a first-born daughter versus a son exhibit higher fertility, shorter birth spacing, and a greater risk of anemia with these outcomes worsening with successive daughters (female births) [23]. Yet, another panel study that followed women aged 15–49 years over six years found no differential impact of having sons compared to daughters on mother's status in the household measured using

freedom of movement, household decision-making, access to bank account, and ownership of assets [24]. Overall, there is mixed empirical evidence on the differential impact of having sons on women's empowerment and status. Finally, no study to date from India has focused on studying son preference and empowerment measures among AGYW.

Extensive global evidence, including from South Asia, suggests that women's empowerment is associated with delayed initiation of childbearing, lower total fertility, greater birth spacing, higher contraceptive usage among women, and increased access to maternal health care [16, 25–27], however, little is known about how marriage and fertility shape adolescent girls' and young women's (AGYW) empowerment. Examining how women's empowerment is linked with maternal and child health outcomes has been of interest to researchers trying to improve gender equality and women's health. We argue that studying women's empowerment should be an end goal because it impacts women's longer-term status and life opportunities.

Much of the literature on the topic of marriage and childbearing is dated. Prior empirical work examining the effects of marriage and childbirth on women's empowerment has been largely cross-sectional, with empowerment being assessed as a static measure. A limitation of this approach is that it does not acknowledge empowerment as a dynamic process in which prior and present life experiences may influence empowerment later. The research on this topic of marriage and childbearing in the South Asian context has been somewhat limited in scope due to the paucity of panel data. Panel data that longitudinally follow AGYW over time allows us to examine life transitions such as marriage and childbearing and its impact on empowerment associated with these life events. It is essential to study the trajectory of AGYW empowerment through their life course especially as they transition into marriage and start their reproductive journey. Given that childbearing occurs mostly early in marriage, its impact on women's empowerment may set the stage for the rest of their life in their marital household. In this study, we overcome the challenges and address these gaps by using a dataset that followed a panel of unmarried and married AGYW over a three-year period. We examined how marriage, childbearing, and specifically bearing a son affects changes in AGYW's empowerment in the early part of married life in India.

## Women's empowerment

Women's empowerment is a multidimensional and complex construct. In this paper, we used Kabeer's definition of women's empowerment: "The expansion of people's ability to make strategic life choices in a context where this ability was previously denied to them" [28]. This definition also fits with the broader concept of empowerment as defined in the World Bank's empowerment sourcebook: "the expansion of choice and action to shape one's life" [29]. In both these definitions, the notion of empowerment stems from the state of disempowerment. Empowerment differs from related concepts of gender equality or women's autonomy through two distinct attributes. The first attribute is the explicit focus on the *processes of change* through which women can achieve greater equality or greater choice. The second attribute is *agency* that puts women as central actors in the process of change [28, 30]. This ability to exercise choice further helps women make second-order choices that may be consequential to their quality of life, health, nutrition, and overall well-being.

## Theoretical framework

We draw upon Kabeer's seminal framework for women's empowerment that underscores women's choice as essential to their power in the household and family [28].

In Kabeer's framework, three interrelated components of empowerment determine women's choices as outlined in *Fig 1*. The first component is resources that enable women to

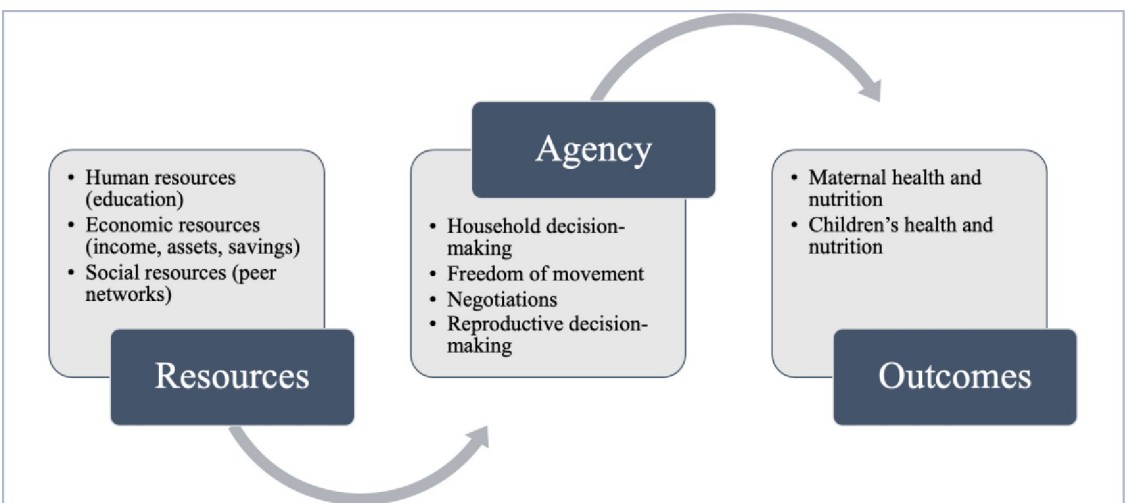

**Fig 1. Women's empowerment conceptualized using Kabeer's framework.**

develop a capacity to articulate their preferences. These include human resources such as educational attainment, life skills, self-efficacy; economic resources such as paid work, earnings, bank accounts, property ownership; and social resources such as access to peer networks, role models, and connections that allow women to improve their situation beyond what is possible alone. The second component of the framework is agency defined as "the ability to define one's goals and act upon them" [28]. Resources help women develop agency that comprises women's ability to make household decisions, freedom to move outside the house, reproductive and sexual decisions, bargaining power, and even cognitive processes such as reflection and analysis [28]. Factors such as education, employment, marriage, and childbearing can enhance or deny women's agency [28]. The third component is achievements that are outcomes achieved through increased access to enabling resources and enhanced agency. Given the interrelatedness of these three components, achievements of a particular moment can be transformed into enhanced resources and agency for a later time, improving women's capacity to make choices in the future [28]. In summary, women's empowerment is a dynamic process, in which women acquire human, social, and economic resources that enable women to gain agency to achieve their desired outcomes or aspirations.

Women's empowerment is not unitary, it has many broad dimensions in women's lives including economic, socio-cultural, political, legal, and psychological empowerment [30]. Based on the measures available in our data, we used economic and socio-cultural empowerment measures in our study across two time points to examine empowerment as a dynamic process. We operationalized economic empowerment using indicators such as bank account ownership, having savings in the bank accounts, and ability to operate bank accounts. In our framework, economic empowerment indicators relate to enabling resources. For socio-cultural empowerment, we relied on two measures of women's empowerment that have been extensively used in the literature—women's participation in household decision-making and freedom of movement [25].

We used a panel dataset that followed unmarried and married AGYW, allowing us to examine how marriage and childbearing shape AGYW's empowerment in the early part of their married life. Using Kabeer's framework, empowerment was examined as a dynamic process over AGYW's life course by studying the immediate changes to empowerment resulting from marriage and attaining motherhood status. Further, given the prevalence of son preference in

the Indian context, we also studied whether having sons conferred additional advantages to AGYW's (i.e., mother's) empowerment compared to having only daughters.

## Research questions

The focus of this paper is on answering the following research questions: Research Question 1 (RQ1): Is marriage associated with a change in empowerment of AGYW? Further, does the relationship between marriage and empowerment depend on whether the household arrangement is patrilocal (newly married women co-residing with in-laws)? Is marriage into a patrilocal household (co-residence with in-laws) associated with a greater change in empowerment for AGYW compared to marriage into a household without in-laws?

Research Question 2 (RQ2): Is motherhood associated with a change in empowerment of married AGYW?

Research Question 3 (RQ3): Is having a son associated with a greater change in empowerment among married AGYW compared to having daughters only?

**Hypotheses.** Our study tests the hypotheses that transition to marriage will be associated with decrease in empowerment among AGYW while transitioning to motherhood will be associated with greater increase in empowerment, and those who bear at least one son will have a greater increase in empowerment.

## Methods

### Data

We used a prospective longitudinal cohort dataset of adolescents from Population Council's UDAYA (Understanding Lives of Adolescents and Young Adults) study, which was conducted in two large northern states—Bihar and Uttar Pradesh (UP)—in India [31]. Two waves of panel data were collected: Wave 1 in 2015–16 and Wave 2 in 2018–19. Wave 1 surveyed 20,594 unmarried boys and girls aged 10–19 and married girls aged 15–19 from a state-representative random sample of 300 urban and rural primary sampling units (PSUs)—villages in rural areas and census wards in urban areas. To answer our study questions, we used a sample of unmarried and married adolescent girls (15–19 years) at Wave 1 and those followed up at Wave 2 when they turned 18–22 years. Of the 7,776 unmarried adolescent girls (15–19 years) interviewed in Wave 1, the team re-interviewed 6,168 AGYW (18–22 years) in Wave 2 with a re-contact rate of 79%. Of the 5,206 married adolescent girls (15–19 years) interviewed in Wave 1, the team re-interviewed 4,257 AGYW (18–22 years) in Wave 2 with a re-contact rate of 82%. The re-contact rates of 79% and 82% in both these samples meet the criteria of minimum acceptable re-contact in longitudinal studies [32].

The UDAYA study was a multi-topic survey that covered different modules related to demographics, economic activity, household work and migration, mass media and social media exposure, social and peer environment, aspirations, gender role attitudes, awareness of sexual and reproductive health matters, empowerment measures, transitions to marriage and parenthood. The study received ethical approval from the Institutional Review Boards of Population Council USA and Centre for Media Studies, India. Since we used de-identified data publicly available from the Harvard Dataverse website, we were exempt from seeking ethical approvals [33]. We did not have access to identifying individual participant data.

Our analytical sample was restricted to the panel of adolescent girls who were unmarried at Wave 1 and followed up at Wave 2 to answer the first research question. Given that we are interested in understanding the relationship between transition to marriage and empowerment, we dropped adolescent girls who were married but had separated or divorced between Wave 1 and Wave 2 (n = 27). We recoded the adolescent girls who were married as unmarried

if they were married but had not had *gauna* (Gauna is a custom in northern India and the ceremony associated with the consummation of marriage often associated with early marriage of girls. Marriage is considered as a ritual union with conjugal life beginning after gauna. Until gauna is performed, married girls stay in their natal home.) (effectively had not moved to their husbands' home to start a marital life) (n = 26) and dropped observations where the household details were not collected at the second wave (n = 24). Our final analytical sample for this question used a balanced panel of 12,130 observations (6,065 observations in each wave).

We used the sample of married adolescent girls who were re-contacted again to answer the second and third research questions. Of the 4,257 adolescent girls surveyed at Wave 1, 257 adolescent girls had still not performed *gauna*. Of the 4,000 adolescent girls who were married and cohabiting with their husbands at Wave 1, 35 were divorced or separated by Wave 2 and dropped observations where the household details were not collected at the second wave (n = 24). Hence, we have a sample of 3,941 adolescent girls (15–19 years) who were already married in 2015–16, still married in 2018–19 as AGYW (18–22 years) and cohabiting with their husbands at both waves. Our paper uses a balanced panel of 7,882 observations (3,941 observations from each wave). We had comparable measures for the same participants across the two waves, including repeated empowerment measures: decision-making, freedom of movement, and access to economic resources.

## Variable definitions and operationalization

### Dependent variables

We used three measures to capture different dimensions of AGYW's empowerment: (i) freedom of movement (or mobility), (ii) decision-making power, both of which reflect AGYW's agency, and (iii) access to economic resources, which captures AGYW's access to enabling resources (Fig 1). The three main dependent variables of empowerment are further detailed below:

1. Freedom of movement or mobility: The survey collected information on whether and how the AGYW travelled to three types of places: (i) a shop or market or a friend/relative's place *inside* their village/ward, (ii) a shop or market or a friend/relative's place *outside* their village/ward, and (iii) to attend any program (health event, girls meeting, sports event) in their village/ward. For each of these questions, we assigned a score of 2 if girls were allowed to go alone, 1 if they were allowed to go but only with a companion, and 0 if they were not allowed to go out at all. Responses were summed to create a scale (0 to 6) with higher responses indicating a higher amount of freedom in mobility.

2. Decision-making power: AGYW were asked who mainly decides about: (i) how much education the respondent should have, (ii) making major household purchases, and (iii) whether the respondent should work or stay at home. For each of these variables, we assigned a score of 2 if the respondent reported making decisions solely, 1 if the respondent takes decisions jointly with others and 0 if the respondent reported having no say in making decisions. Items were summed to form a scale (0 to 6) with higher scores indicating greater decision-making ability of the respondent.

3. Access to economic resources: Three survey questions were included on whether girls had their own bank account (*yes = 1, no = 0*), whether they operated their own account (*yes = 1, no = 0)*, and whether they had savings in their bank accounts (*yes = 1, no = 0)*. Responses were summed to create a scale (0 to 3) with higher responses indicating a greater access to economic resources.

### Independent variables

1. Marriage: An indicator variable for whether the adolescent girl is married or unmarried as captured by the question to ascertain the adolescent girl's marital status (Sample = adolescent girls (15–19 years) who were unmarried at wave 1).

2. Motherhood: An indicator variable for whether the adolescent girl has experienced a live childbirth (Sample = adolescent girls (15–19 years) who were married at wave 1).

3. Having a son: A categorical variable to indicate whether the AGYW has had daughters only, at least one son, or no children (reference category). Sample was adolescent girls (15–19 years) who were married at wave 1.

### Covariates

Sociodemographic covariates that may be related to marriage, fertility, and empowerment were selected *a priori* based on prior literature to ensure that sociodemographic variables are not confounding the explanations for primary hypotheses. Past literature suggests that adolescent girls' education is associated with an increase in their empowerment, status, and intra-household bargaining power [24, 34]. Following Jensen's panel study which found that the introduction of satellite television was significantly associated with an increase in women's household decision-making and a decrease in her son preference, we included exposure to mass media as one of the controls [35]. The role of household-level factors such as a household wealth index, the presence of in-laws in the marital household, and family type (nuclear versus joint household) have also been associated with women's empowerment [7, 36–38]. Household wealth index was computed using principal components analysis with a standard set of questions on a household's ownership of consumer items such as a television, scooter and other assets, dwelling characteristics such as flooring material, type of drinking water source, and toilet facilities.

To understand the association between the transition to marriage and changes in empowerment, we controlled for time-varying covariates such as adolescent girls' education (as a continuous score) and exposure to mass media (television, radio, and newspaper). We also controlled for whether those AGYW who were married by the second wave had already *borne a child* to disentangle the independent effects of being married on changes in empowerment. To study the heterogeneous effects of the association between marriage and changes in empowerment, we included co-residence with in-laws as an effect modifier. The effect modifier, *co-residence with in-laws*, was only asked in the second wave of the UDAYA data. AGYW who were married into a household with in-laws were coded as 1, and 0 otherwise. Co-residence variable was coded as 0 in wave 1 for all adolescent girls and as 0 in wave 2 for AGYW who remained unmarried, so the coefficients can be interpreted as interaction with marital status. Finally, we also controlled for survey wave (Wave 2) because of the possibility of secular trends in empowerment over time.

### Analysis

We used fixed effects modeling, specifically modeling the within-individual variation across the two survey waves, to examine how changes in an adolescent girls' marital status and AGYW's motherhood status are associated with changes in their empowerment. A key strength of this approach is the ability to control for unobserved confounders at the individual level that do not change over time, removing much of the omitted variable bias. In our analyses, the "marriage effect" can be interpreted as the effect of marriage on changes in AGYW

empowerment over time. Similarly, the "motherhood effect" can be interpreted as the effect of becoming a mother on changes in AGYW empowerment over time and the "mother of son effect" can be interpreted as the effect of bearing a son compared to having daughters only or no children on changes in AGYW empowerment over time.

First, we generated descriptive statistics for all variables of interest, by wave of data collection, that allowed us to explore the characteristics of 15–19 year-old unmarried and married adolescent girls at both waves.

For the first question, we used individual linear fixed-effects regressions to capture how a change in marital status from being unmarried to married is associated with changes in the empowerment domains mentioned previously using the specification below.

$$Y_{it} = \beta_1 + \beta_2\, X_{it} + \beta_3\, X_{it} \times I_{it} + \boldsymbol{\beta_4}\, \boldsymbol{Z_{it}} + \theta_t + \alpha_i + \varepsilon_{it}$$

Here $Y_{it}$ represents one of the three dependent variables including intrahousehold decision-making power, freedom of movement, and access to economic resources of adolescent $i$ at time $t$ *(all adolescent girls who were unmarried at t = 1)* for three separate regression models;

$X_{it}$ represents the marital status of AGYW $i$ at time $t$ *(all adolescent girls were unmarried at t = 1)* for those married into a household without in-laws;

$X_{it} \times I_{it}$ takes the value 1 if *adolescent girls* gets married into a household with in-laws at the second wave *(t = 2)* and 0 otherwise for an AGYW $i$ and represents the interaction term between marriage and co-residence with in-laws;

$\boldsymbol{Z_{it}}$ is a vector of time-varying AGYW-level characteristics such as educational attainment and mass-media exposure;

$\theta_t$ is a fixed effect for time (operationalized via a dummy variable for the second wave);

$\alpha_i$ is a fixed subject-specific effect for AGYW i;

$\varepsilon_{it}$ is a random error term that follows a normal distribution with mean 0 and constant variance.

We also used the *lincom* command to calculate the effect of marriage on empowerment domains for the interaction term on those co-residing with in-laws compared to not co-residing with their in-laws.

To estimate the association between the transition to motherhood and becoming mothers of son and changes in empowerment, we ran similar individual linear fixed-effects regressions on the sample of married AGYW for each of the domains of empowerment over time listed previously (Please refer to Supplementary Material (S1 File) for model specifications).

Across all the models, we set the level of statistical significance (Wald p-value) at 0.05. Robust standard errors were used to relax assumptions of constant variance and constant within-subject residual correlations. All data were analyzed using Stata Version 15.1 [39].

## Results

### Characteristics of the sample

*Table 1* (the first column) shows the demographic characteristics of unmarried adolescent girls (n = 6,065) at Wave 1 across both states. At Wave 1, on average, unmarried adolescent girls were 16 years old. Approximately 6% of adolescent girls never received formal education, a little over half (55.9%) of adolescent girls completed some level of secondary education, and 32% of the adolescent girls completed some primary education. Less than one in five adolescent girls (18%) reported having undertaken paid work in the 12 months preceding the survey. Over half (55%) the adolescent girls lived in rural settings. A majority (72.9%) of adolescent girls reported being exposed to mass media, including radio, TV, and newspapers, at least once a month. One in five adolescent girls (21%) reported ever having received sexual and

**Table 1. Characteristics of unmarried and married adolescent girls interviewed at Wave 1 (2015–16), UDAYA longitudinal survey.**

| | n (%) of AGYW-unmarried (n = 6,065) at Wave 1, 2015–16 | n (%) of AGYW-married (n = 3,941) at Wave 1, 2015–16 |
|---|---|---|
| **Age (mean ± SD)** | 16.6 ± 1.3 | 18.0 ± 1.0 |
| **Education** | | |
| No formal education | 384 (6.3%) | 1,079 (27.3%) |
| Completed some primary education (1–8 Grade) | 1,945 (32.0%) | 1,480 (37.6%) |
| Completed some secondary education (9–12 Grade) | 3,387 (55.9%) | 1,263 (32.1%) |
| Completed some college education | 349 (5.8%) | 119 (3.0%) |
| **Household wealth index** | | |
| Poorest | 1,525 (25.1%) | 991 (25.1%) |
| Poor | 1,524 (25.1%) | 981 (24.9%) |
| Rich | 1,525 (25.1%) | 993 (25.2%) |
| Richest | 1,491 (24.7%) | 976 (24.7%) |
| **Religion** | | |
| Hindu | 4,479 (73.9%) | 3,336 (84.6%) |
| Muslim | 1,562 (25.7%) | 594 (15.0%) |
| Other | 24 (0.4%) | 11 (0.2%) |
| **Caste** | | |
| Scheduled Caste / Scheduled Tribe | 1,262 (20.8%) | 1,181 (29.9%) |
| Other Backward Caste | 3,444 (56.8%) | 2,375 (60.2%) |
| General Caste | 1,359(22.4%) | 385 (9.7%) |
| **Recent paid work** | | |
| No | 4,975 (82.1%) | 3,516 (89.3%) |
| Yes | 1,090 (17.9%) | 425 (10.7%) |
| **Household size (mean ± SD)** | 6.5 ± 2.7 | 5.8 ± 3.0 |
| **Exposure to mass media (TV, radio, newspaper) at least once a month** | | |
| No | 1,652 (27.2%) | 2,018 (51.2%) |
| Yes | 4,413 (72.7%) | 1,923 (48.8%) |
| **Received sexual reproductive health education** | | |
| No | 4,741 (78.1%) | 3,512 (89.1%) |
| Yes | 1,324 (21.9%) | 429 (10.9%) |
| **Residence** | | |
| Urban | 2,729 (45.0%) | 1,428 (36.2%) |
| Rural | 3,336 (55.0%) | 2,513 (63.7%) |
| **State** | | |
| Uttar Pradesh (UP) | 3,315 (54.6%) | 1,299 (32.9%) |
| Bihar | 2,750 (45.4%) | 2,642 (67.1%) |
| **Marital Status at Wave 1** | | |
| Unmarried | 6,065 | - |
| Married | - | 3,941 |
| **Had a childbirth at Wave 1** | | |
| No | 6,065 | 2,128 (54.0%) |
| Yes | - | 1,813 (46.0%) |

reproductive health information. Of the 6,065 unmarried adolescent girls, 24.4% (n = 1482) got married in the interval between Wave 1 and Wave 2.

The second column of *Table 1* describes the demographics of married adolescent girls at Wave 1 (n = 3941). At Wave 1, the average age of married adolescent girls is 18 years. Unlike the cohort of unmarried adolescent girls, a little over a quarter (27%) of the married adolescent girls did not receive any formal education, 37.6% completed some primary education and about a third (32.1%) completed some secondary education. Only one in ten adolescent girls (10.7%) reported doing paid work recently in the 12 months preceding the survey. Sixty-three percent of married adolescent girls lived in rural settings. Less than half (48.8%) of married adolescent girls reported being exposed to mass media, including radio, TV, and newspapers, at least once a month. One in ten adolescent girls (10.9%) reported ever having received sexual and reproductive health information. Of the 3,941 married adolescent girls, 46% had begun childbearing by the first Wave of the survey.

Below we summarize the regression results.

## Transition to marriage and empowerment domains

*Table 2* presents the coefficients from the individual panel linear fixed effects regression. Controlling for having a childbirth by wave 2, adolescent girls' education, exposure to mass media, and time, we found that being married is associated with a reduction in mobility of AGYW ($\hat{\beta}$ = -0.45, 95%CI = -0.63, -0.27, p<0.001) for AGYW not marrying into household where they co-reside with in-laws. Co-residence with in-laws was a significant effect modifier of the relationship between marriage and freedom of movement ($\hat{\beta}$ = -0.87, 95%CI = -1.06, -0.69, p<0.001) holding other covariates constant. Specifically, AGYW who marry into a household with in-laws had a more pronounced decrease in freedom of movement compared to AGYW who married and did not co-reside with their in-laws ($\hat{\beta}$ = -1.13, 95%CI = -1.47, -1.20, p<0.001).

On average being married was associated with greater decision-making power ($\hat{\beta}$ = 0.29, 95%CI: 0.10, 0.47, p = 0.002) for AGYW who did not marry into a household where they co-reside with their in-laws, holding all else constant. Marrying into a household with in-laws is an effect modifier in the relationship between marriage and decision-making power ($\hat{\beta}$ = -0.39, 95%CI: -0.57, -0.21, p = 0.002). Specifically, we did not observe a statistically significant association between marriage and decision-making power among AGYW who marry into a household with in-laws compared to AGYW who were married without co-residence with in-laws ($\hat{\beta}$ = -0.10, 95%CI: -0.25, 0.04, p = 0.174).

We also found that on average, there was a positive association between transition to marriage and increased access to economic resources ($\hat{\beta}$ = 0.37, 95%CI = 0.23, 0.51, p<0.000) for AGYW not marrying into household where they co-reside with in-laws. We did not find that the association between marriage and access to economic resources varied by living arrangement (co-residence with in-laws).

## Association between transition to motherhood and empowerment domains

*Table 3* depicts the results from individual fixed effects regressions examining the association of motherhood status and empowerment domains among the panel of married AGYW. Becoming a mother is associated with an increase in mobility of AGYW ($\hat{\beta}$ = 0.12, 95% CI = 0.02, 0.23, p = 0.023). We did not find a positive association between becoming a mother and intrahousehold decision-making power of AGYW ($\hat{\beta}$ = 0.09, 95%CI = -0.00, 0.19,

**Table 2. Regression coefficients from individual fixed effects regressions for the association between marriage and freedom of movement, decision-making power, access to economic resources among adolescent girls and young women (AGYW) in the two study states (n = 12130).**

| Marriage and empowerment | | | | | | | | | |
|---|---|---|---|---|---|---|---|---|---|
| | Freedom of movement | | | Decision-making power | | | Access to economic resources | | |
| | Coeff. | 95%CI | p-value | Coeff. | 95%CI | p-value | Coeff. | 95%CI | p-value |
| **Being married** | -0.45 | -0.63, -0.27 | <0.000 | 0.29 | 0.10, 0.47 | 0.002 | 0.37 | 0.23, 0.51 | <0.000 |
| **Marriage X co-residence with in-laws (Ref: Co-residence with husband/other adults)** | | | | | | | | | |
| Being married and co-residing with in-laws | -0.87 | -1.06, -0.69 | <0.000 | -0.39 | -0.57, -0.21 | <0.001 | -0.08 | -0.22, 0.06 | 0.256 |
| **Controls** | | | | | | | | | |
| Having had a childbirth | 0.15 | -0.02, 0.31 | 0.086 | -0.02 | -0.18, 0.14 | 0.805 | 0.28 | 0.16, 0.41 | <0.001 |
| Educational attainment of AGYW | 0.02 | -0.01, 0.05 | 0.133 | -0.06 | -0.09, -0.02 | <0.001 | 0.10 | 0.07, 0.12 | <0.001 |
| Exposure to mass media | 0.10 | 0.02, 0.18 | 0.015 | 0.01 | -0.07, 0.09 | 0.874 | 0.06 | 0.00, 0.13 | 0.037 |
| Wave (Ref: Wave 1) | | | | | | | | | |
| Wave 2 | 0.32 | 0.26, 0.39 | <0.000 | 0.08 | 0.01, 0.15 | 0.026 | 0.21 | 0.16, 0.26 | <0.001 |
| Mean (SD) for unmarried AGYW | 4.6 (1.2) | | | 1.9 (1.5) | | | 1.7 (1.1) | | |
| **Number of Observations** | 12130 | | | 12130 | | | 12130 | | |

Notes: [1] Husband/wife alone or with wife' parents or other relatives; Robust 95 per cent confidence intervals

p = 0.058) controlling for other variables and time (wave) in the model. Further, becoming a mother was also positively correlated with higher access to economic resources ($\hat{\beta}$ = 0.24, 95% CI = 0.16, 0.32, p < 0.001) holding other variables constant.

## Estimating the role of having sons on empowerment among married AGYW

Based on the models below in Table 4, holding all controls and time constant, we find evidence that having daughters only (compared with no children) is positively associated with access to

**Table 3. Regression coefficients from three individual fixed effects regressions for the association between motherhood and freedom of movement, decision-making power, and access to economic resources among married AGYW in India (n = 7882).**

| Motherhood and empowerment | | | | | | | | | |
|---|---|---|---|---|---|---|---|---|---|
| | Freedom of movement | | | Decision-making power | | | Access to economic resources | | |
| | Coeff. | 95%CI | p | Coeff. | 95%CI | p | Coeff. | 95%CI | p |
| **Motherhood** | 0.12 | 0.02, 0.23 | 0.023 | 0.09 | -0.00, 0.19 | 0.058 | 0.24 | 0.16, 0.32 | <0.001 |
| **Controls** | | | | | | | | | |
| **Girls' education** | -0.05 | -0.14, 0.04 | 0.271 | -0.04 | -0.13, 0.05 | 0.361 | -0.19 | -0.26, -0.12 | <0.001 |
| **Wealth index scores** | -0.03 | -0.06, 0.00 | 0.076 | -0.04 | -0.07, -0.01 | 0.004 | 0.03 | 0.00, 0.05 | 0.027 |
| **Household size** | -0.03 | -0.04, 0.01 | 0.002 | -0.04 | -0.05, -0.02 | <0.001 | 0.00 | -0.02, 0.01 | 0.599 |
| **Mass media exposure** | 0.09 | 0.00, 0.19 | 0.048 | 0.09 | 0.00, 0.17 | 0.048 | 0.17 | 0.10, 0.24 | <0.001 |
| **Wave 2 (Ref: Wave 1)** | 0.80 | 0.73, 0.87 | <0.001 | 0.11 | 0.04, 0.17 | 0.001 | 0.55 | 0.49, 0.61 | <0.001 |
| **Average for girls without children** | 3.2 (1.3) | | | 1.6 (1.4) | | | 1.2 (0.9) | | |
| **Number of Observations** | 7882 | | | 7882 | | | 7882 | | |

Note: Robust 95 per cent confidence intervals.

**Table 4. Regression coefficients from three separate individual fixed effects regressions for the association between sex composition of children and freedom of movement, decision-making power, and access to economic resources among adolescent girls and young women.**

| Mother of sons and empowerment | | | | | | | | | |
|---|---|---|---|---|---|---|---|---|---|
| | Freedom of movement | | | Decision-making power | | | Access to economic resources | | |
| | Coeff. | 95%CI | p-value | Coeff. | 95%CI | p-value | Coeff. | 95%CI | p-value |
| **Having daughters** (Ref: no children) | 0.10 | -0.02, 0.22 | 0.094 | 0.10 | -0.01, 0.20 | 0.075 | 0.23 | 0.14, 0.31 | <0.001 |
| **Having at least one son** (Ref: no children) | 0.15 | 0.02, 0.27 | 0.020 | 0.09 | -0.03, 0.20 | 0.134 | 0.25 | 0.16, 0.34 | <0.001 |
| **Additional effect of having a son compared to daughters only** | 0.05 | -0.07, 0.17 | 0.432 | -.010 | -0.11, 0.09 | 0.848 | 0.024 | -0.07, 0.11 | 0.588 |
| **Controls** | | | | | | | | | |
| **Girls' education** | -0.05 | -0.14, 0.04 | 0.287 | -0.04 | -0.13, 0.05 | 0.357 | -0.19 | -0.25, -0.12 | <0.001 |
| **Wealth index** | -0.03 | -0.06, 0.00 | 0.078 | -0.04 | -0.07, -0.01 | 0.003 | 0.03 | 0.00, 0.05 | 0.026 |
| **Household size** | -0.03 | -0.04, 0.01 | 0.002 | -0.04 | -0.05, -0.02 | <0.001 | 0.00 | -0.02, 0.01 | 0.588 |
| **Mass media exposure** | 0.10 | 0.00, 0.19 | 0.047 | 0.09 | 0.00, 0.17 | 0.048 | 0.17 | 0.10, 0.25 | <0.001 |
| **Wave 2 (Ref: Wave 1)** | 0.79 | 0.72, 0.87 | <0.001 | 0.11 | 0.04, 0.18 | 0.001 | 0.55 | 0.49, 0.60 | <0.001 |
| **Number of Observations** | 7882 | | | 7882 | | | 7882 | | |

Note: Robust 95 per cent confidence intervals.

economic resources. Similarly, having at least one son (compared with no children) is associated with greater freedom of movement and greater access to economic resources of AGYW. However, there is no evidence of an additional effect of having a son over and above the effect of having daughters. In other words, there is no differential change in empowerment between AGYW who have borne at least one son compared to having only daughters.

## Discussion

To our knowledge, this is the first panel study of adolescent girls and young women (AGYW) from India to explore the changes in their social and economic empowerment associated with critical life events such as marriage, motherhood, and son preference. By leveraging data from unmarried and married adolescent girls across two time points, we examined empowerment as a dynamic process, overcoming a limitation of prior cross-sectional studies that examine empowerment as a static measure. Drawing on Kabeer's seminal framework, we examined how transition to marriage and motherhood are associated with measures of AGYW's agency and enabling resources. Specifically, we tested whether marrying into a patrilocal household (i.e., co-residence with in-laws) moderates the association between marriage and empowerment. Finally, we examined the effects of motherhood on empowerment and studied the effects of becoming a mother of a son over having only daughters (or no daughters) using the married AGYW dataset.

As hypothesized, our results showed that on average, marriage was associated with lower freedom of movement among AGYW controlling for covariates and time. We also found that married AGYW co-residing with their in-laws had a much lower freedom of movement compared to AGYW who do not co-reside with their in-laws. These results are consistent with prior evidence from India suggesting that co-residence with the mother-in-law was negatively associated with the daughter-in-law's mobility [7, 40, 41],ability to form social connections [7], and use of family planning [42, 43]. Living with in-laws means more hierarchy within the household that tends to decrease the empowerment of the youngest daughters-in-law [7, 44, 45]. Newly married women (daughters-in-law) often have the lowest status in the household compared to men and other senior women in the household especially during the early period of their marriage [4, 5]. For instance, daughters-in-law are often expected to follow gendered

norms of eating last in the household and are socially isolated from their peer networks [7, 46, 47]. Apprehensions about the use of contraception, influence of outside peers on daughters-in-law, and fear of losing authority can potentially explain the negative effects on mobility of daughters-in-law [7].

We found that transition to marriage was associated with greater decision-making power for AGYW who did not co-reside with their in-laws. However, we did not observe a statistically significant association between transition to marriage and decision-making power among those who co-resided with their in-laws compared to those who did not co-reside with their in-laws. This finding contradicts results from an Indian cross-sectional study that noted that compared to women married into nuclear families, women ages 15–49 years married in extended family households living with in-laws had lower decision-making power [40, 41].

As hypothesized, we found that transition to motherhood was positively correlated with greater freedom of movement, marginally higher intrahousehold decision-making power, and better access to economic resources. Our results are consistent with prior evidence from another panel study from India that found that motherhood was associated with increased freedom of movement and gaining access to a bank account [24]. Another study using retrospective data from a northern state in India also demonstrated that women in childbearing stages had higher agency and less fear of spousal violence than those who had not yet started childbearing [48].

Our results of increased empowerment among young AGYW are not entirely surprising because women's status is inherently tied to their fertility. However, this increase in women's empowerment may come at the expense of in-laws' pressures to bear children immediately after marriage, potentially reflecting the limited ability of women to control their fertility [49]. In an ethnographic study of young Nepali women, some young women reported feeling uncomfortable asserting their agency out of a desire to conform to being a "good woman and daughter-in-law" [50]. Such forms of "strategizing" by women in the 'classic patriarchal belt' (from North Africa and Middle East to South Asia) has been given the term "patriarchal bargain", in which young brides accept "hardship" as a tradeoff to gain economic security through their spouse [6]. Our results also complement the research carried out in other contexts where women's value is linked to their fertility such as the Middle East [51]. In addition to the norms surrounding fertility, childbearing brings with it practices surrounding childcare that might offer the women opportunities to step outside their homes, seek healthcare for their children, and make decisions within the household. One interpretation for increased access to economic resources could be due to conditional cash transfer programs being targeted at pregnant women to increase institutional births, expanding women's access to bank accounts and savings [52].

Contrary to our hypothesis, we did not find statistically significant evidence that having at least one son compared to having daughters only (or no daughters) conferred additional changes in girls' freedom of movement, intrahousehold decision-making power, and access to economic resources. But our study findings are consistent with several of the recent studies on this topic of son preference. Overall, these findings are consistent with recent findings from another panel study that examined motherhood effects on a panel of Indian women (15–49 years) that did not find that motherhood effects on empowerment measures were different by sex of child [24]. In fact, another study from Bangladesh and India found that having daughters improved mother's decision-making power and freedom of mobility compared to having sons [53]. Between 1990 and 2021, son preference in India declined from 40% to 18% likely due to increase in female education and exposure to television that have been associated with greater gender-equitable attitudes [54]. Furthermore, another recent study from Bangladesh

also noted a greater desire for gender balance between children due to increased female education, female employment, and decline in joint family living situations [55].

However, our results deviated from two cross-sectional studies in which authors found that having more sons was associated with greater autonomy among women in Uttar Pradesh, India [36] and having a male first birth was associated with clean cooking fuels that authors assert could be due to greater intrahousehold bargaining power of women or elevated social status for women in urban India [56]. One plausible explanation for non-differential changes in AGYW's empowerment by sex composition could be progressive gender policies and national girl child focused programs such as *Beti Bachao Beti Padhao (*Save and Educate the Girl Child) that have been found to improve sex ratio at birth in one of the Indian states with the worst sex ratio at birth [57, 58]. Further, there has been an improvement in sex ratio at birth in both the study states (Bihar and UP), potentially reflecting a broader societal change in attitudes towards girl children [9]. Finally, since the participants in our study are young and may continue childbearing, more rounds of data collection in the future may allow us to continue examining differential effects on the trajectory of empowerment by sex composition of children.

Despite making some significant contributions to the literature, we acknowledge that our study has some limitations. First, we studied only two northern states in India which are similar in social and cultural norms surrounding women's status and role in the society and thus our results are not generalizable to other regions of India. Second, the data were collected in two waves in 2015–16 and 2018–19. This three-year duration between the two waves could possibly be insufficient to capture the longer-term effects of marriage and childbearing on domains of empowerment including mobility, intrahousehold decision-making, and access to economic resources from a lifecourse perspective. While we were able to study immediate changes in empowerment following marriage and childbirth, we were not able to study lifetime influences on the process of empowerment. Further, the empowerment measures we used capture only the economic and socio-cultural dimensions of empowerment and do not capture other dimensions of empowerment such as political, legal, and psychological empowerment. Given the multidimensional nature of empowerment, empowerment in one domain may not necessarily translate into empowerment in other domains [30, 59, 60]. The analysis was also limited in its ability to unpack the mechanisms through which marriage and childbirth are associated with changes in empowerment over time.

Our study has many strengths worth noting. To our knowledge, this is one of the two studies in the Indian context that has examined the relationship between childbirth and empowerment and the first study that has longitudinally analyzed how the transition from being unmarried adolescent girls to married AGYW impacts AGYW's empowerment over time. Second, we were able to utilize this panel dataset that followed state-representative samples of unmarried and married AGYW at two time points to examine the process of women's empowerment immediately after their marriage and childbirth. In India, childbearing follows soon after marriage, hence by following recently married young women, we were able to study short-term changes in empowerment over time. We also were able to examine empowerment as a process over time by using panel fixed effects that remove the omitted variable bias arising from unobserved time-invariant variables. We also grounded our analysis in one of the seminal frameworks in the field of empowerment, whose underuse has been one of the drawbacks of empirical studies in the field of women's empowerment literature [60]. By using within-person fixed effects analysis, any measurement error related to different cognitive understanding or semantic meaning of empowerment questions across individuals have been minimized [61]. Third, our measures of empowerment are relevant to the context of North Indian settings and are also like the indicators measured in Demographic Health Surveys rendering it

comparable. Finally, by controlling for wave (or time), we were able to control for secular trends and maturity the adolescents may acquire over age and time in their marriage.

## Policy implications

Achieving gender equality and empowering all women and girls is the fifth goal in the 2030 agenda for the Sustainable Development Goals (SDGs) [62]. Our findings on marriage and empowerment highlight the importance of understanding vulnerabilities associated with being newly married during adolescence, particularly how limited freedom of movement and lower decision-making power could impede newly married adolescents' access to essential services, including family planning and other health-seeking behaviors. Our findings also highlight the need to have health interventions that target both daughters-in-law and mothers-in-law together. This is particularly important because in the Indian context childbearing follows immediately after marriage. Our findings also place importance on changing restrictive gendered cultural norms that put newly married women at the lowest status in the marital household. Empowering women has benefits not only for themselves, but also for their children as women's empowerment has been associated with many maternal and child health outcomes [16].

Childbearing was associated with greater empowerment across the dimensions chosen for the study. Findings underscore the high value placed on childbearing in this setting and re-emphasize the need to address harmful social and gender norms that define women's status within the confines of their reproductive potential. Helping shift restrictive gender norms so that the multitude of other contributions (female labor force participation, unpaid caregiving) that women make are more highly valued is essential.

Currently the research on this topic has been limited by the availability of panel datasets in India. Recommendations for future research include designing and collecting panel data that follows the AGYW (15–24 years) population over an extended period, similar to the National Longitudinal Study of Adolescent to Adult Health [63] in the United States, which gathers rich demographic, social, familial, socioeconomic, behavioral, psychosocial, cognitive, and health survey data from participants and their parents. With recent changes to sex selective abortion in India, the patterns of son preference are declining, and there has been a notable rise in gender-equitable fertility preferences. However, such topics can benefit from having regular detailed panel datasets. Changes in sex ratio also require Census data that has not been collected in India since 2010. Availability of more recent data will allow researchers to examine trends in marriage, fertility, sex ratio, and how these are associated with women's empowerment and agency.

## Conclusion

In this study, we examined the association between transition to marriage and childbirth on social and economic dimensions of empowerment using a panel data from two large states of India. We extended the existing literature by investigating adolescent girls' empowerment over time and whether marriage, motherhood, and bearing at least one son are associated with multiple types of AGYW empowerment in India. Transitioning into a marital union was associated with reduced freedom of movement for AGYW, particularly among those living with their in-laws. However, for AGYW not residing with in-laws, marriage was correlated with increased influence in decision-making. Becoming a mother was associated with increased freedom of movement, slightly enhanced authority within the household's decisions, and improved access to economic opportunities. Our study also did not observe statistically significant evidence of having at least one son compared to having daughters only (or no daughters)

resulted in changes to AGYW's freedom of movement, intrahousehold decision-making power, and access to economic resources. Findings emphasize the need for interventions that target newly married young women along with key household members such as mothers-in-law to empower them. Taken together, our findings reiterate the notion that empowerment is a dynamic process impacted by life events such as marriage and childbirth.

## Supporting information

**S1 Checklist.** *PLOS ONE* **clinical studies checklist.**
(DOCX)

**S2 Checklist. STROBE statement—checklist of items that should be included in reports of observational studies.**
(DOCX)

**S1 File. Supplementary information-model specifications.**
(DOCX)

## Author Contributions

**Conceptualization:** Lakshmi Gopalakrishnan.

**Formal analysis:** Lakshmi Gopalakrishnan.

**Methodology:** Lakshmi Gopalakrishnan, Sophia Rabe-Hesketh.

**Supervision:** Stefano Bertozzi, Sophia Rabe-Hesketh.

**Validation:** Sophia Rabe-Hesketh.

**Writing – original draft:** Lakshmi Gopalakrishnan.

**Writing – review & editing:** Lakshmi Gopalakrishnan, Stefano Bertozzi, Sophia Rabe-Hesketh.

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
