## [Decision Letter · Decision Letter 0]

3 Aug 2023

PONE-D-23-10607Role of marriage, motherhood, son preference on adolescent girls’ empowerment: Evidence from a panel study in IndiaPLOS ONE

Dear Ms. Lakshmi Gopalakrishnan ,

Thank you for submitting your manuscript to PLOS ONE. After careful consideration, we feel that it has merit but does not fully meet PLOS ONE’s publication criteria as it currently stands. Therefore, we invite you to submit a revised version of the manuscript that addresses the points raised during the review process.

Try and pay a close attention to the following issues as you revise your manuscriptSeparate the research questions from the hypothesesTry to use more recent literature as most of the literature cited are very oldNote that your conclusion did not reiterate the effect of son preference on empowerment as a strong norm in India. Try and work on this.Include recommendations for future researchPlease submit your revised manuscript by Sep 17 2023 11:59PM If you will need more time than this to complete your revisions, please reply to this message or contact the journal office at plosone@plos.org. Please include the following items when submitting your revised manuscript:A rebuttal letter that responds to each point raised by the academic editor and reviewer(s). You should upload this letter as a separate file labeled 'Response to Reviewers'.A marked-up copy of your manuscript that highlights changes made to the original version. You should upload this as a separate file labeled 'Revised Manuscript with Track Changes'.An unmarked version of your revised paper without tracked changes. You should upload this as a separate file labeled 'Manuscript'.If applicable, we recommend that you deposit your laboratory protocols in protocols.io to enhance the reproducibility of your results. Protocols.io assigns your protocol its own identifier (DOI) so that it can be cited independently in the future. For instructions see: https://journals.plos.org/plosone/s/submission-guidelines#loc-laboratory-protocols. Additionally, PLOS ONE offers an option for publishing peer-reviewed Lab Protocol articles, which describe protocols hosted on protocols.io. Read more information on sharing protocols at https://plos.org/protocols?utm_medium=editorial-email&utm_source=authorletters&utm_campaign=protocols.

We look forward to receiving your revised manuscript.

Kind regards,

Olufunmilayo Olufunmilola Banjo, Ph.D

Academic Editor

PLOS ONE

Journal Requirements:

Reviewers' comments:

Reviewer's Responses to Questions

**Comments to the Author**

1. Is the manuscript technically sound, and do the data support the conclusions?

Reviewer #1: Yes

Reviewer #2: Yes

2. Has the statistical analysis been performed appropriately and rigorously? 

Reviewer #1: Yes

Reviewer #2: Yes

3. Have the authors made all data underlying the findings in their manuscript fully available?

Reviewer #1: Yes

Reviewer #2: Yes

4. Is the manuscript presented in an intelligible fashion and written in standard English?

Reviewer #1: Yes

Reviewer #2: Yes

5. Review Comments to the Author

Reviewer #1: Summary of the report

Using two waves of panel datasets conducted in India, a longitudinal study followed both married and unmarried girls and young women in India. The authors argued that changes in the usual environment of young girls and women and the cultural norms surrounding preference for sons affect women’s overall life course and also shape the level of empowerment across the course of their lives in their patrilocal homes. It further highlighted the effect on the health and well-being of these adolescent girls and young women in India with reference to other studies that indicated negative outcomes of practice of son’s preference resulting in the short spacing of births, higher fertility, and prevalence of anaemia with accompanied effect on the health and wellbeing of mothers. They also explored whether having sons conferred additional advantages to AGYW's (i.e., mother’s) empowerment compared to having only daughters. The authors concluded that empowerment is a dynamic process and has an impact on life events such as marriage and childbirth.

Comments

• The title of the article is clear, however, the reference to only adolescent girls excludes participants over the age of 19, making the term unsuitable. (Adolescents are persons between the ages of 10 and 19 years, while the age range 18 -22 falls within WHO's definition of young people, which refers to individuals between ages 10 and 24).

• The abstract is of proper length (290 characters) and has all the sections included.

• The quality and quantity of data, quality, and adequacy of the statistical techniques

o The authors used secondary data, which I believe is dependable and adequate,

o The method employed described the data used for the survey, how it was obtained, and when and where they were completed. The section described the participants, instrumentation, and procedures was outlined. The controls and sample sizes are appropriate and the methodology is detailed enough to be reproduced by other researcher

o A data analysis subsection was provided in the methods section using STATA 15.1 the Statistical Software for analysis was reported.

o The statistical test used was described, most of the findings were reported and the pertinent frequencies as well as the coefficient, confidence interval, and p-values were included in the text.

o Actual p-values were reported, set at a level of statistical significance (Wald p-value) at 0.05. Likewise, the standard error was used to indicate the variability of a data set. Confidence intervals were also reported with the effect sizes (regression coefficients) in the regression analyses.

o With respect to the results, the tables were clearly presented and correctly labelled. Data presentation was done in a meaningful, understandable way, and with ease of interpretation.

○

• The adequacy, relevance, and recency of the literature cited.

o More than 80% of the literature cited is not recent, the majority are above 10 years.

• The adequacy and relevance of the data generated and

o the main features of important characteristics of the participants were displayed in the table generated i.e socio-economic characteristics of the respondents, the total number, age, gender, marital status, freedom of movement or mobility, decision-making power, access to the economy, and so on.

o the tables and figures were well prepared with a clear title, the row, and each column helped to clarify the different data presented in the table.

o the tables and figures provided the total number of participants or the number of participants in each subgroup.

o statistical summary statistics, tests, and methods in tables and figures were named.

• Quality of discussion with respect to the research problem(s)

o The authors introduced the problem statement under study, and the relevant background information was discussed which led to the specific purpose of the study. There was also a statement of the research question and hypotheses.

o The authors discussed the purpose of the research & the hypothesis, in relation to the theoretical framework proposed and to previous literature on the topic. Attempts were made by the authors to examine, interpret, and qualify the results from the study. Although this research has contributed to the literature on the role of marriage, motherhood, and son preference on adolescent girls’ empowerment, no recommendation was made for future research in this area. To a very large extent, the summary statement was based on appropriate evidence.

Specific areas for improvement

• The research questions and corresponding hypotheses should be separated. Hypotheses should have a sub-heading of their own separate from the research questions.

• Recommendations for future research in this area be included.

• The authors should update the citations to more recent literature.

• The referencing style is in MLA format, I guess. If the answer is yes, the authors should ensure that the title of the journal is in an inverted comma and the name of the journal is italicized.

For example,

o Harris-Fry H, Shrestha N, Costello A, Saville NM. “Determinants of intra-household food allocation between adults in South Asia – a systematic review”. Int J Equity Health. 2017 Jun 21;16(1):107.

o Do M, Kurimoto N. “Women’s empowerment and choice of contraceptive methods in selected African countries”. Int Perspect Sex Reprod Health. 2012 Mar;38(1):23–33.

• On pages 15 & 16, the authors mentioned the inclusion of interaction terms in the model the analysis.

Overall, the article was interesting to read but a bit heavy to read and understand. Meanwhile, particularly interesting in the article, is the discussion on empowerment using Kabeer’s framework as the theoretical approach. The framework was employed in the definition of women’s empowerment to answer the questions brought up by the authors. Using a prospective longitudinal cohort dataset of adolescent girls and young women collected in two waves from 2015-19 from two large northern states, Bihar and Uttar Pradesh (UP) in India. The findings from the study fit the article’s purpose, however, their conclusion did not reiterate the effect of son preference on empowerment as this is a strong cultural norm in India shown to impact the lives of women.

Reviewer #2: The author needs to take a second look at the introduction section by bring into the work some statistical result in terms of percentages.

Summary Review: Role of marriage, motherhood, son preference on adolescent girls’ empowerment: Evidence from a panel study in India

1. Importance of article/Relevance and Appeal to national / international scholarly Excellent Good Moderate Poor

2. Title of the study Excellent Good Moderate Poor

3. Original and Independent Research Excellent Good Moderate Poor

4. Presentation and readability Excellent Good Moderate Poor

5. Statement of problem(s)/aim(s)/objective(s) Excellent Good Moderate Poor Not Available

6. Literature review Excellent Good Moderate Poor Not Available

7. Appropriateness of (if applicable)

7.1. Research plan and design

7.2. Data presentation/Discussion

7.3. Conclusion/Recommendations

Excellent Good Moderate Poor

Excellent Good Moderate Poor

Excellent

Excellent

Good

Good

Moderate

Moderate

Poor

Poor

8. To what extent is the line of argumentation in the article clear, cohesive and logical? Excellent Good Moderate Poor

9. Contribution to theory Excellent Good Moderate Poor

10. Contribution to practice Excellent Good Moderate Poor

NOTE:

• This paper will be better situated if it follows the chronological order of writing an Introduction.

6. PLOS authors have the option to publish the peer review history of their article (what does this mean?). If published, this will include your full peer review and any attached files.

Reviewer #1: **Yes: **Omolayo Oluwatope

Reviewer #2: No

---

## [Author Response · Author response to Decision Letter 0]

14 Aug 2023

We are deeply grateful to the editor and reviewers for their feedback and comments. We believe this has improved the quality of work we set out to do. We have provided responses as well as made edits to the manuscript (in track changes). Where possible below, we clearly marked the page numbers with the changes made. Our responses are provided below. Thanks! 

Editor Comments

• Separate the research questions from the hypotheses

Thank you. I have made the edits you have suggested on Page 10 and 11. 

• Try to use more recent literature as most of the literature cited are very old.

A lot of the old papers are seminal ones, but I have taken your suggestion into account. Yes, throughout the paper I have added the recent literature. Main changes have been made in the introduction (Page 3-6) and discussion (Page 25). 

• Note that your conclusion did not reiterate the effect of son preference on empowerment as a strong norm in India. Try and work on this.

I checked my analysis and found the same results. My conclusion remains the same. I found several recent papers that also have similar findings of reducing son preference (Page 25 and 29).

Include recommendations for future research

This is an excellent recommendation, I have incorporated this on Page 29

Reviewer 1 comments:

• The title of the article is clear, however, the reference to only adolescent girls excludes participants over the age of 19, making the term unsuitable. (Adolescents are persons between the ages of 10 and 19 years, while the age range 18 -22 falls within WHO's definition of young people, which refers to individuals between ages 10 and 24).

I edited the title.

• The adequacy, relevance, and recency of the literature cited. More than 80% of the literature cited is not recent, the majority are above 10 years. 

This has been edited and new literature has been added.

Specific areas for improvement

• The research questions and corresponding hypotheses should be separated. Hypotheses should have a sub-heading of their own separate from the research questions.

Done 

• Recommendations for future research in this area be included.

Thank you, noted and Done

• The authors should update the citations to more recent literature.

Thank you, noted and done. But the old papers are seminal pieces of work and I have retained them. I added new citations but the challenge is very limited research have been done on this topic with AGYW population. 

• The referencing style is in MLA format, I guess. If the answer is yes, the authors should ensure that the title of the journal is in an inverted comma and the name of the journal is italicized.

Referencing style is Vancouver, as prescribed by PLOSONE. I have not made any changes to this.

Overall, the article was interesting to read but a bit heavy to read and understand. Meanwhile, particularly interesting in the article, is the discussion on empowerment using Kabeer’s framework as the theoretical approach. The framework was employed in the definition of women’s empowerment to answer the questions brought up by the authors. Using a prospective longitudinal cohort dataset of adolescent girls and young women collected in two waves from 2015-19 from two large northern states, Bihar and Uttar Pradesh (UP) in India. The findings from the study fit the article’s purpose, however, their conclusion did not reiterate the effect of son preference on empowerment as this is a strong cultural norm in India shown to impact the lives of women.

I did my best to reorganize the introduction to make it more palatable and readable for the audience and emphasized the conclusion of not finding an effect of son preference on empowerment.

---

## [Editor Report · Decision Letter 1]

25 Aug 2023

PONE-D-23-10607R1Role of marriage, motherhood, son preference on adolescent girls’ and young women's empowerment: Evidence from a panel study in IndiaPLOS ONE

Dear Dr. Gopalakrishnan,

Thank you for submitting your manuscript to PLOS ONE. After careful consideration, we feel that it has merit but does not fully meet PLOS ONE’s publication criteria as it currently stands. Therefore, we invite you to submit a revised version of the manuscript that addresses the points raised during the review process

We look forward to receiving your revised manuscript.

Kind regards,

Olufunmilayo Olufunmilola Banjo, Ph.D

Academic Editor

PLOS ONE

Journal Requirements:

A**dditional Editor Comments:** .Therefore, we invite you to revise the manuscript again by paying close attention to the following issues:You may not need to state hypotheses based on research questions, since you did not present your results based on the stated hypotheses. Rather, you can just state one single hypothesis that captures the aim of the studyRevision of the title is not sufficient...there is the need to pay close attention to other issues to ensure a smooth flow from the title throughout the manuscript. For example, there is the need to give clearer explanation as regarding the use of the 2 waves of data. Let it be clear that it is only the adolescent girls you considered in the first wave, while you considered both adolescents and young women in the second wave, due to age progression from the first to the second wave. There is the need to further revise the Methods and Analysis sections to remove unnecessary repetitions, that make the methods section clumsy to understand. 

---

## [Author Response · Author response to Decision Letter 1]

26 Aug 2023

We thank the Editor for his careful review and comments. Please find below our responses. Thank you.

• You may not need to state hypotheses based on research questions, since you did not present your results based on the stated hypotheses. Rather, you can just state one single hypothesis that captures the aim of the study

I deleted the hypotheses and summarized it in a single sentence as recommended by the Editor.

• Revision of the title is not sufficient...there is the need to pay close attention to other issues to ensure a smooth flow from the title throughout the manuscript. For example, there is the need to give clearer explanation as regarding the use of the 2 waves of data. Let it be clear that it is only the adolescent girls you considered in the first wave, while you considered both adolescents and young women in the second wave, due to age progression from the first to the second wave. 

o I have revised the title a little bit further. Also, throughout the paper, where reference to Wave 1 is being discussed, I changed it to adolescent girls. And for Wave 2 related discussion, I use AGYW – to encompass the age ranges of both adolescent girls and young women (AGYW). 

• There is the need to further revise the Methods and Analysis sections to remove unnecessary repetitions, that make the methods section clumsy to understand. 

o We are sorry the methods section was verbose making it hard to read. I have now amended the methods and analysis section to make it concise. I moved the estimating equations to Supplementary Material. Please refer to pages 17-19 for edits (in track changes).

---

## [Editor Report · Decision Letter 2]

12 Sep 2023

Role of marriage, motherhood, son preference on adolescent girls’ and young women’s empowerment: Evidence from a panel study in India

PONE-D-23-10607R2

Dear Dr. Lakshmi Gopalakrishnan,

We’re pleased to inform you that your manuscript has been judged scientifically suitable for publication and will be formally accepted for publication once it meets all outstanding technical requirements.

Kind regards,

Olufunmilayo Olufunmilola Banjo, Ph.D

Academic Editor

PLOS ONE
---

## [Editor Report · Acceptance letter]

19 Sep 2023

PONE-D-23-10607R2 

Role of marriage, motherhood, son preference on adolescent girls’ and young women’s empowerment: Evidence from a panel study in India 

Dear Dr. Gopalakrishnan:

I'm pleased to inform you that your manuscript has been deemed suitable for publication in PLOS ONE. Congratulations! Your manuscript is now with our production department. 

Kind regards, 

on behalf of

Dr. Olufunmilayo Olufunmilola Banjo 

Academic Editor

PLOS ONE